# Synthesis of Novel Halogenated Heterocycles Based on *o*-Phenylenediamine and Their Interactions with the Catalytic Subunit of Protein Kinase CK2

**DOI:** 10.3390/molecules26113163

**Published:** 2021-05-25

**Authors:** Maria Winiewska-Szajewska, Agnieszka Monika Maciejewska, Elżbieta Speina, Jarosław Poznański, Daniel Paprocki

**Affiliations:** Institute of Biochemistry and Biophysics, Polish Academy of Sciences, Pawińskiego 5a, 02-106 Warsaw, Poland; mwin@ibb.waw.pl (M.W.-S.); agniesza@ibb.waw.pl (A.M.M.); elasp@ibb.waw.pl (E.S.)

**Keywords:** kinase CK2, differential scanning fluorimetry, molecular modeling, halogenated heterocycles, hydrophobic contribution, ligand binding, inhibitory activity, cell toxicity

## Abstract

Protein kinase CK2 is a highly pleiotropic protein kinase capable of phosphorylating hundreds of protein substrates. It is involved in numerous cellular functions, including cell viability, apoptosis, cell proliferation and survival, angiogenesis, or ER-stress response. As CK2 activity is found perturbed in many pathological states, including cancers, it becomes an attractive target for the pharma. A large number of low-mass ATP-competitive inhibitors have already been developed, the majority of them halogenated. We tested the binding of six series of halogenated heterocyclic ligands derived from the commercially available 4,5-dihalo-benzene-1,2-diamines. These ligand series were selected to enable the separation of the scaffold effect from the hydrophobic interactions attributed directly to the presence of halogen atoms. In silico molecular docking was initially applied to test the capability of each ligand for binding at the ATP-binding site of CK2. HPLC-derived ligand hydrophobicity data are compared with the binding affinity assessed by low-volume differential scanning fluorimetry (nanoDSF). We identified three promising ligand scaffolds, two of which have not yet been described as CK2 inhibitors but may lead to potent CK2 kinase inhibitors. The inhibitory activity against CK2α and toxicity against four reference cell lines have been determined for eight compounds identified as the most promising in nanoDSF assay.

## 1. Introduction

Protein kinase CK2 is considered as one of the essential human proteins, which is involved in numerous cellular processes. This is a highly pleiotropic kinase, which can phosphorylate hundreds of protein substrates including transcriptional factors, proteins affecting the structure of DNA/ RNA, and/or involved in RNA synthesis and translation, signaling proteins and also pathogenic virus replication machinery proteins [1]. The numerous natural substrates of kinase CK2 make it indispensable for many cellular functions including cell viability, apoptosis, cell proliferation and survival, angiogenesis, DNA damage and repair, the ER-stress response, the regulation of carbohydrate metabolism, and in the nervous system [2]. Kinase CK2 is the holoenzyme that acts in vivo as a heterotetramer, consisting of two catalytic (α- and/or α’) and two regulatory (β) subunits, but also may act as a monomeric kinase alone, the regulatory subunit only improving the activity [3]. 

Since kinase CK2 was found to be overexpressed in many forms of human cancers [4], its role in cancer pathogenesis is thoroughly investigated. CK2 could promote cell survival by phosphorylation of sites adjacent to caspase cleavage sites, which blocks its activity [4,5,6,7]. Kinase CK2 is also involved in DNA damage response and DNA repair pathways, which may additionally contribute to the regulation of cancer cell survival [8]. Despite the fact that the exact function of hCK2 in oncogenesis remains unknown and also its contribution may be different among various carcinomas [9,10,11], widespread interest has been expressed in targeting this kinase therapeutically [12].

Dozens of kinase CK2 inhibitors have been already described. Among them, ATP-competitive inhibitors, halogenated benzotriazole, and benzimidazole derivatives play an important role. Following 5,6-dichloro-1-(β-D-ribofuranosyl)benzimidazole (DRB) [13], 4,5,6,7-tetrabromo-1*H*-benzotriazole (TBBt) and 4,5,6,7-tetrabromo-1*H*-benzimidazole (TBBz) were identified as strong and selective kinase CK2 inhibitors [14]. Further studies on these compounds resulted in the synthesis of numerous potent kinase CK2 inhibitors, including 4,5,6,7-tetrabromo-2-(dimethylamino)benzimidazole (DMAT) [15]. Besides, studies also showed that derivatives of 4,5,6,7-tetraiodo-1*H*-benzimidazole (TIBz) inhibit α and α′ subunits of CK2 much more than TBBz does [16]. Sulfurated derivatives of TBBz are also efficient kinase CK2 inhibitors, e.g., 4,5,6,7-tetrabromo-1*H*-benzoimidazole-2(3H)-thione (K22) and 4,5,6,7-tetrabromo-2-(methylthio)-1*H*- benzoimidazole (K37) inhibit kinase stronger than TBBt [17,18]. 5,6,7,8-tetrabromo-1-methyl-2,3-dihydro-1*H*-benzoimidazo[1,2-*a*] imidazole (K44), containing an additional imidazolidine ring in its structure, was also reported to strongly inhibit CK2 [17,18]. Structures of all these CK2 inhibitors are shown in Figure 1.

The majority of halogenated ligands target the ATP-binding site of a protein kinase; however, their poses differ significantly [19,20,21,22,23]. Recent studies have indicated that the binding mode of halogenated benzotriazoles and benzimidazoles is generally driven by a competition between a hydrogen bond or salt bridge and halogen bonding [20], while ligand hydrophobicity contributes significantly to the binding affinity [23,24]. Systematic structural studies on complexes of halogenated ligands pointed the preferred geometry [25,26,27] and topology [21,28] of a halogen bond in protein-ligand systems, however, the thermodynamic contribution of a halogen bond to free energy of ligand binding remains debatable, being estimated in the range of 0.2 [29] to 5 kcal/mol [30].

Our long-term systematic studies on protein kinase CK2 ligands have demonstrated that halogenated benzotriazole derivatives showed moderate inhibitory activity against CK2, which is closely related to the physicochemical properties of the ligand [31,32,33,34,35,36]. We have also analyzed the effect of substitution pattern, demonstrating that benzotriazole derivatives carrying halogen atoms at positions 5 and 6 interact with CK2 much stronger than those substituted at positions proximal to the triazole ring (positions 4 and 7) [34,35]. What is more, the presence of the iodine atom in the structure significantly enhances the affinity [37], which is in line with results obtained by Janeczko et al. for massively iodinated benzimidazole derivatives [16]. Both benzotriazoles and benzimidazoles possess *o*-phenylenediamine fragment in their structure (see Figure 1). Looking for the novel groups of compounds as putative kinase CK2 inhibitors, we decided to synthesize groups of similar compounds, based on *o*-phenylenediamine configuration.

Here, we synthesized and described six series of heterocyclic halogenated ligands derived from 4,5-dihalogeno-benzene-1,2-diamines. These ligands were selected to enable the separation of the scaffold effect from the hydrophobic interactions attributed directly to the halogen atoms. We measured the hydrophobicity using reverse-phase HPLC. The binding affinities to the catalytic subunit of human protein kinase CK2 (hCK2α) were initially assessed using low-volume differential scanning fluorimetry (nanoDSF), and the dominating modes of binding were analyzed using a molecular modeling approach. Cytotoxicity against four reference cell lines was tested for the most promising compounds along with their inhibitory activity against CK2.

## 2. Results

### 2.1. Chemistry

We synthesized six series of halogenated heterocyclic compounds starting from 4,5-dihalogenobenzene-1,2-diamines (**1a**–**e**). While the fluorinated, chlorinated, and brominated amines are commercially available, 4,5-diiodobenzene-1,2-diamine was synthesized, according to the protocol described previously, using a two-step procedure starting from 1,2-dinitrobenzene [37]. 2,1,3-Benzothiadiazole derivatives (**2a**–**e**) were synthesized from the corresponding diamines during reaction with thionyl chloride and triethylamine performed in dichloromethane [38]. 2-Trifluoromethyl-1*H*-benzimidazole derivatives (**3a**–**e**) were synthesized by heating corresponding amine **1a**–**e** in trifluoroacetic acid in the presence of the catalytic amount of conc. HCl [39]. 2-hydroxymethyl-1*H*-benzimidazole derivatives (**4a**,**c**–**e**) were obtained by heating diamine with 2-hydroxyacetic acid in hydrogen chloride solution [40]. 1*H*-benzo[d]imidazol-2(3*H*)-ones (**5a**,**c**–**e**) were synthesized from **1a**–**e** and *N*,*N*′-carbonyldiimidazole (CDI) mixed in DMF [41]. Quinoxaline derivatives (**6a**,**c**–**e**) were obtained by heating substrate **1a**–**e** with glyoxal solution in ethanol. Quinoxaline-2,3-diol derivatives (**7a**–**e**) were obtained by heating diamine **1a**–**e** with oxalic acid in DMF. The synthesis strategy of compounds **2**–**7** is summarized in Scheme 1.

### 2.2. Hydrophobicity Data

RP-HPLC method, a fast and efficient alternative for standard logD determination in octan-1-ol/water system [42], was used to assess the hydrophobicity of all synthesized compounds. This method has been successfully used in our laboratory for halogenated benzotriazole derivatives, including the evaluation of the effect of hydrophobicity on their binding to the catalytic subunit of protein kinase CK2 [24,43].

As expected, the substitution of two hydrogen atoms with the larger halogen atoms: fluorine, chlorine, bromine, or iodine, respectively, always results in an increase of apparent hydrophobicity (Figure 2A). This trend is generally independent on the scaffold of the molecule (Figure 2B).

Indeed, regardless of the scaffold, the increase of hydrophobicity upon halogen substitution in distant positions of the benzene ring of 2-trifluoromethyl- 1*H*- benzimidazole (**3a**), 2-hydroxymethyl-1*H*-benzimidazole (**4a**), and 1*H*-benzo[d]imidazol-2(3*H*)-one (**5a**) is identical. A similar trend is also observed in the case of quinoxaline-2,3-diol (**7a**) and benzo-1,2-diamine (**1a**), however, for the latter, the effect of halogenation is shifted downwards by 0.1 log(τ) units. The largest changes in hydrophobicity experience quinoxaline (**6a**) derivatives, which should be directly attributed to the lack of the effect of electronegative halogen atom on the proximal proton-donating groups. Such effect, partially compensating halogen hydrophobicity, was evidenced in protein-ligand structures, in which hydrogen bonds proximal to a halogen atom were statistically shortened (so must be regarded as stronger) when a nitrogen atom close to the halogen atom (less than six chemical bonds) was a hydrogen bond donor [44]. No such configuration is present in **6a**, so the effect of halogen substitution is larger because it cannot be compensated in **6c**–**e** by the halogen-induced strengthening of solute-solvent interaction. Finally, the effect of halogen substitution in the most hydrophobic 2,1,3-benzothiadiazole (**2a**) remains irregular, which putatively results from an uncontrolled aggregation of these solutes.

### 2.3. Binding Affinity—Thermal Shift Assay

Binding affinities of the newly synthesized compounds towards hCK2α were assessed with the low-volume differential scanning fluorimetry (nanoDSF). The shift of the temperature of thermal denaturation in the presence of 10-fold excess of a particular ligand relative to the *apo* form of hCK2α (ΔT_m_) can be used as a semi-qualitative measure of the binding affinity [35] or inhibitory activity [45].

Groups of ligands with different scaffolds vary visibly in their affinity to the target protein (see Table 1 and Figure 3). Interestingly, two of the most hydrophobic groups of compounds, 2,1,3-benzothiadiazoles (**2a**–**e**) and quinoxalines (**6a**–**e**), virtually do not interact with hCK2α—for all these compounds, ΔT_m_ value never exceeds 1 °C. Benzene-1,2-diamine derivatives (**1a**–**e**) also interact weakly with hCK2α (ΔT_m_ value 1.7 °C for **1e**), but this effect should be attributed directly to the size of the ligand—the absence of the second aromatic ring in the scaffold that appears to be necessary for the efficient ligand-binding to the kinase [20]. There is no general correlation between hydrophobicity and binding affinity, however, when one analyzes individual series representing the same scaffold. A comparison of log(τ) and ΔT_m_ for the protein–ligand complexes indicates that any increase in ligand hydrophobicity is clearly reflected by an increase in binding affinity, as long as the ligand binds to the target (see Figure 3). Moreover, it seems that the effect of Cl, Br, and I substitution on the stabilization of the protein–ligand complexes not only does not depend on the structure of the heterocyclic ring but is directly proportional to the hydrophobicity of the halogen. The latter is evidenced by almost parallel lines representing the contribution of halogen atoms in individual ligands. It could be thus anticipated that for the studied series of ligands, the thermodynamic contribution of non-specific hydrophobic interactions of halogen atoms to the free energy of ligand binding remains separated from that of the direct electrostatic interactions involving a more polar heterocyclic ring. This hypothesis was tested with in silico modeling of protein–ligand structures.

### 2.4. Inhibitory Activity

Since in the thermal shift assay, most of the studied compounds seem to have a negligible affinity towards hCK2α, only eight compounds were further subjected to the direct measurement of IC_50_—solely compounds with the thermal shift above 2 °C may be considered as at least moderate inhibitors of hCK2α.

Comparison of thermal shift assay data (∆Tm) with the directly measured IC_50_ confirms that nanoDSF is an attractive method for inhibitors screening, however, for ligands with different scaffolds, hence different modes of binding, this might not be directly reflected in the biochemical assay.

Finally, it should be mentioned that the inhibitory activity of two compounds found the most promising according to nanoDSF (i.e., 5,6-diiodo-2-trifluoromethyl-1*H*- benzimidazole, **3e**, and 6,7-diiodooquinoxaline-1,3-diol,**7e**) also exhibit the highest inhibitory activity with micromolar IC_50_ values (see Table 2).

### 2.5. Cell Toxicity

The same four bromo and four iodo- derivatives that, according to nanoDSF assay, interact with the catalytic subunit of CK2α (scaffolds 3, 4, 5, 7) were tested against the reference cancer cell lines: epidermoid carcinoma A-431, colorectal carcinoma HCT116, HCT116p53^−/−^, and normal fibroblasts BJ. All of them displayed moderate activity to reduce cell viability (<100 µM). Interestingly, compound **3e** was identified as even more active than **3d**, the promising activity of which has been already reported [46]. In this study compound, **3d** displayed IC_80_ values between 2 and 29.1 μM tested for five cancer cell lines (IM-9, MOLT3, U-937, MCF-7, and PC-3), and 8.7 μM on phytohemagglutinin-stimulated normal human lymphocytes. It was especially active against two cell lines: MCF-7 (breast carcinoma) and PC-3 (prostate carcinoma), with IC_80_ values below 3 μM. Our data show the selectivity index of 3**e** ~10 (relative to BJ fibroblasts), which makes this compound worth further investigations. We were unable to determine the cytotoxicity for **5e**, for which the observed nonmonotonic dose-response effect was most probably caused by uncontrolled aggregation. All cell toxicity data are summarized in Table 2, and the analyses of viability are shown in Appendix A. The toxicity is generally uncorrelated with the inhibitory activity; however, this effect can be directly assigned to membrane permeability. This hypothesis is strongly supported by a clear correlation between compound hydrophobicity (log (τ)) and IC_50_ values estimated from viability data (Figure 4).

Interestingly, contrary to the enzymatic assay, the activity of **7e** remains marginal (IC_50_ ~100 µM), but such a low value should be assigned to a moderate hydrophobicity of this compound. In this context, one can expect that esterification of the two hydroxyl groups of **7e** would improve its apparent activity in viability tests.

### 2.6. Molecular Modeling

Molecular modeling was initially performed to assess whether a given scaffold could be harbored at the ATP binding site. Free energy of ligand binding (ΔG_bind_) was estimated with the aid of the VINA-AutoDock algorithm implemented in Yasara Structure [47]. The obtained ensembles of structures agreed qualitatively with our knowledge concerning the binding of halogenated ligands at the ATP-binding site of CK2α [21,48,49]. Thus, all chlorinated, brominated, and iodinated ligands preferably adopt the orientation, in which both halogen atoms are solvent-protected, being close to the hinge region, while the heterocyclic ring points towards the polar region of the ATP-binding site formed by sidechains of Lys68, Glu81, and Asp175. The competition between hydrophobic and electrostatic interactions is visible for the majority of tested ligands, again clearly identifying the balance of hydrophobic and electrostatic interactions as the main driving force.

Interaction between a fluorinated ligand (**1b**, **2b**, **3b**, **7b**) and the protein is found generally overestimated (Figure 5, red points). Contrarily, the binding affinity of the majority of iodinated and some brominated ligands is underestimated (Figure 5, blue and green points). However, the latter effect may reflect the absence of the parameterization of halogen bonding interactions in the classical force-field used in the scoring function [50]. On the other hand, these discrepancies can be treated as indicators for the existence of halogen bonding in these complexes, which mainly applies to **3e**, **4e**, and **7e**. Interestingly, despite the discrepancy in the estimated binding affinity, most of the low-energy poses of 5,6-diiodo-, 5,6-dibromo, and 5,6-dichloro- derivatives preserve halogen atoms pointing toward the hinge region, thus, in the orientation, enabling halogen-bonding, with the heterocyclic ring close to Lys68 (see Appendix A).

## 3. Discussion

We have synthesized six groups of halogenated heterocycles, based on benzene-1,2,-diamine derivatives. To analyze their applicability as the precursors of potential hCK2 inhibitors, we used molecular modeling together with the nanoDSF method, which enabled qualitative comparison of their binding affinity. Many studies have been carried out on CK2 Type I inhibitors and some preferences of ligands in CK2 ATP-binding sites are already well known [20,33,35,48,51]. Besides the obvious steric hindrance of the scaffold, the two main driving forces of binding ligand at the ATP- binding site of CK2α are clearly identified: the electrostatic contribution together with hydrophobic effect [20,33,35]. Negatively charged ligands commonly bind deeper in the CK2α pocket, preferably interacting with Lys68 [20]. Additionally, the substitution with hydrophobic halogen atoms significantly increases the affinity towards the CK2α by a favorable change in the physico-chemical properties of the ligand. The latter also includes the possible formation of halogen bonds with the hinge region [48].

Using these different groups of compounds, we analyzed the interplay between the scaffold and the effect of substitution with halogen atoms. For smaller scaffolds, like benzotriazole/benzimidazole, unsubstituted on triazole/imidazole ring, a salt bridge or hydrogen bond formation with Lys68 and halogen bonding with the hinge region are largely mutually exclusive because these two regions are in the hCK2α, too far apart for these interactions to occur simultaneously [20,48]. This is why we tested larger scaffolds with either a larger heterocyclic ring (**6a**–**e**, **7a**–**e**) or substitution in the imidazole, both of them to preserve possible interactions with a hinge. We analyzed solely dihalogenated compounds of the same pattern of halogen atoms at the benzene ring to assure that predominating interactions with the hinge region (occurring with more halogen atoms) and configuration effect (depending on the halogenation pattern) will not predominate or interfere with the interdependence we were analyzing [24,34,43]. Interestingly, for the tested halogen configuration, we can notice that only ligands with uneven charge distribution on the non-benzene ring (groups 3, 4, 5, 7) exhibit measurable affinity towards CK2, while ligands even much hydrophobic but without proton-donating groups (series 2 and 6) do not stabilize the complex (i.e., ΔT_m_ < 1 °C). This observation indicates that two halogen atoms are insufficient to stabilize the ligand in the complex with a position close to the linker region, and the formation of hydrogen bonds with the lysine seems to be necessary. Such dependence of ligand position on the number of halogen atoms was observed for the halogenated benzotriazoles, where 5,6 dibromobenzotriazole interacts with Lys68 (pdb 6TLP), while tetrabromobenzotriazole interacts either with the hinge region or with Lys68 (pdb 6TLL). What is more, an inspection of in silico determined structures for complexes with ligands that bind to the protein (ΔT_m_ < 1 °C) points out to this unique interplay, as ligands with no halogen atom or with fluorine (which is not capable for the formation of halogen bond [52]) display in almost all cases different orientation relative to the hinge region (see Appendix A), possibly forming hydrogen bonds. The substitution with other halogen atoms switches the preferable orientation of the ligand, which has an effect that is somehow explicable by hydrophobic interactions. However, since the force-field used in the docking procedure does not include the explicit potential for a halogen bond, these interactions are underestimated.

These different orientations of chlorinated, brominated, and iodinated compounds might explain why the relation of measured log(τ) and ΔT_m_ values for series representing the same compound scaffold is linear solely for these three substituents, but not for hydrogen and fluorine (see Figure 3).

Six out of eight compounds with the highest shift in nanoDSF assay displayed at least moderate inhibitory activity in the enzymatic assay; among them, **3e** and **7e** were the most active in both experiments. Some of tested scaffolds have already been reported in the literature. Thus, 1*H*-benzo[d]imidazol-2(3*H*)-one derivatives (**5a**–**d**) moderately interact with CK2 (ΔT_m_ values are up to 2.4 °C for iodinated derivative with and IC_50_ = 13 ± 2 μM, while 4,5,6,7-tetrabromo-1*H*-benzo[d]imidazol-2(3*H*)-one inhibits CK2 stronger (IC_50_ = 0.39 μM and K_i_ = 0.20 μM) [15]. 2-Trifluoromethyl-1*H*-benzimidazole derivatives have been also reported: compound **3d** with IC_50_ value against CK2 of 28 μM (being in line with measurement by us, 15 ± 3 μM) and 4,5,6-tribrominated and tetrabrominated analogs with the inhibitory activity of 1.2 μM and 0.6 μM, respectively [53]. Replacement of bromine atoms with iodine additionally improves the activity up to 0.12 μM [54].

It is worth noting that for these two 5,6-halogenated scaffolds, the observed tendency is opposite to that observed for their tetrahalogenated analogs. Thus, 2-trifluoromethyl- 1*H*-benzimidazole interacts stronger than the 1*H*-benzo[d]imidazol-2(3*H*)-one counterpart (7 vs. 13 μM). This again points out the subtle interplay between the scaffold and the substitution with halogen atoms, indicating that the interaction of 2-trifluoromethyl-1*H*- benzimidazole with Lys68 is slightly stronger than for its isostructural analog-carrying carbonyl group.

In this context, quinoxaline-2,3-diol derivatives (**7d**,**e**) might lead to novel potent CK2 inhibitors, as both inhibit hCK2α comparable to 2-trifluoromethyl-1*H*-benzimidazole derivatives (**3d,e**), stabilizing the complex for approx. 4 °C (IC_50_ = 13 and 2 vs. 15 and 7 µM, respectively), but have not been reported as kinase CK2 inhibitors yet.

Finally, the viability tests performed for reference cell lines showed moderate activity for some compounds. Interestingly, IC_50_ values determined in this test are correlated with a compound’s hydrophobicity rather than the inhibitory activity in the CK2α assay. This indicates a dominant role of membrane permeability, but it may also indicate that CK2α is not a major target for studied test compounds.

## 4. Materials and Methods

### 4.1. Chemistry

All starting materials and solvents for reactions were purchased from Sigma Aldrich (now Merck KGaA, Darmstadt, Germany), Fluorochem (Hadfield, UK), ABCR (Karlsruhe, Germany), or Chempur (Piekary Śląskie, Poland). ^1^H-NMR spectra were recorded with Varian 500 MHz spectrometer. TMS or the residual solvent signal were used as the internal standard. High-resolution mass spectrometry spectra were recorded on a TQ OrbitrapVelos instrument (Thermo Scientific, Waltham, MA, USA). The reaction progress was monitored by the thin-layer chromatography analysis using silica gel plates (Kieselgel 60F_254_. Merck, Darmstadt, Germany). Column chromatography was performed on Silica Gel 60 (0.040–0.063 mm. Merck, Darmstadt, Germany).

Detailed experimental data, as well as spectral data for all synthesized compounds, are available in the Appendix A.


*Synthesis of 2,1,3-benzothiadiazole derivatives (**2a–e**)*


Benzene-1,2-diamine derivative (5 mmol) was dissolved in dry DCM (15 mL). Triethylamine (3 mL) was added, then thionyl chloride (15 mmol, 1.1 mL) dissolved in 3 mL of dry DCM was added dropwise at 0 °C. The reaction mixture was allowed to reach room temperature, then it was heated at 40 °C overnight. Afterward, the reaction mixture was filtered. The filtrate was evaporated, the crude product was purified by column chromatography on silica gel using DCM as eluent.

2,1,3-benzothiadiazole (**2a**) Yield = 66%. **^1^**H-NMR (400 MHz, d-DMSO) δ 7.62–7.72 (2H, m, Ar*H*), 7.99–8.80 (2H, m, Ar*H*); **^13^**C-NMR (100 MHz, d-DMSO) δ 121.24, 129.83, 154.19; **HRMS** calcd. for C_6_H_5_N_2_S [M + H]^+^: 137.01680, found: 137.01673.

5,6-difluoro-2,1,3-benzothiadiazole (**2b**) Yield = 66%. **^1^**H-NMR (500 MHz, d-DMSO) δ 8.18–8.27 (2H, m, Ar*H*); **^13^**C-NMR (125 MHz, d-DMSO) δ 106.37, 106.54, 150.46, 1651.77, 151.94, 153.81, 153.97; **HRMS** calcd. for C_6_H_3_F_2_N_2_S [M + H]^+^: 172.99795, found: 172.99803.

5,6-dichloro-2,1,3-benzothiadiazole (**2c**) Yield = 75%. **^1^**H-NMR (500 MHz, d-DMSO) δ 8.53 (2H, s, Ar*H*); **^13^**C-NMR (125 MHz, d-DMSO) δ 121.75, 133.54, 152.61; **HRMS** calcd. for C_6_H_3_Cl_2_N_2_S [M + H]^+^: 204.93885, found: 204.93872.

5,6-dibromo-2,1,3-benzothiadiazole (**2d**) Yield = 78%. **^1^**H-NMR (500 MHz, d-DMSO) δ 8.67 (2H, s, Ar*H*); **^13^**C-NMR (125 MHz, d-DMSO) δ 124.95, 126.26, 153.32; **HRMS** calcd. for C_6_H_3_Br_2_N_2_S [M + H]^+^: 292.83782, found: 292.83741.

5,6-diiodo-2,1,3-benzothiadiazole (**2e**) Yield = 47%. **^1^**H-NMR (500 MHz, d-DMSO) δ 8.79 (2H, s, Ar*H*); **^13^**C-NMR (125 MHz, d-DMSO) δ 111.39, 130.38, 154.22; **HRMS** calcd. for C_6_H_3_I_2_N_2_S [M + H]^+^: 388.81008, found: 388.30973.


*Synthesis of 2-trifluoromethylo-1H-benzimidazole derivatives (**3a–e**)*


Benzene-1,2-diamine derivative (2 mmol) was dissolved in trifluoroacetic acid (2 mL) and a catalytic amount of concentrated HCl was added. The reaction mixture was heated at reflux overnight. The reaction was quenched by the addition of 50 mL of concentrated NaHCO_3_ solution; afterwards, it was extracted by ethyl acetate (3 × 30 mL). Combined organic layers were dried by MgSO_4_, then the solvent was evaporated. The crude product was purified by column chromatography on silica gel using 85:15 (hexane:AcOEt) as eluent (compounds **3d** and **3e**) or by recrystallization from hexane (compounds **3a**, **3b**, and **3c**).

2-trifluoromethylo-1*H*-benzimidazole (**3a**) Yield = 73%. **^1^**H-NMR (500 MHz, d-DMSO) δ 7.34–7.42(2H, m, Ar*H*), 7.68-7.77(2H, m, Ar*H*); **^13^**C-NMR (125 MHz, d-DMSO) δ 115.88, 118.02, 120.17, 122.32, 124.09, 139.90, 140.22; **HRMS** calcd. for C_8_H_6_F_3_N_2_ [M + H]^+^: 187.04776, found: 187.04773.

5,6-difluoro-2-trifluoromethylo-1*H*-benzimidazole (**3b**) Yield = 67%. **^1^**H-NMR (500 MHz, d-DMSO) δ 7.71–7.85 (2H, m, Ar*H*); **HRMS** calcd. for C_8_H_4_F_5_N_2_ [M + H]^+^: 223.02892, found: 223.02882.

5,6-dichlorlo-2-trifluoromethylo-1*H*-benzimidazole (**3c**) Yield = 85%. **^1^**H-NMR (500 MHz, d-DMSO) δ 8.00 (2H, s, Ar*H*); **HRMS** calcd. for C_8_H_4_F_3_Cl_2_N_2_ [M + H]^+^: 254.96981, found: 254.96976.

5,6-dibromo-2-trifluoromethylo-1*H*-benzimidazole (**3d**) Yield = 40%. **^1^**H-NMR (500 MHz, d-DMSO) δ 8.16 (2H, s, Ar*H*); **^13^**C-NMR (125 MHz, d-DMSO) δ 115.88; **HRMS** calcd. for C_8_H_4_F_3_Br_2_N_2_ [M + H]^+^: 342.86878, found: 342.86896.

5,6-diiodo-2-trifluoromethylo-1*H*-benzimidazole (**3e**) Yield = 33%. **^1^**H-NMR (500 MHz, d-DMSO) δ 8.32 (2H, m, Ar*H*); **HRMS** calcd. for C_8_H_4_F_3_I_2_N_2_ [M + H]^+^: 438.84105, found: 438.84123.


*Synthesis of 2-hydroxymethylo-1H-benzimidazole derivatives (**4a**,**c–e**)*


Benzene-1,2-diamine derivative (5 mmol) and hydroxyacetic acid (15 mmol, 1.14 g) were dissolved in water (1.5 mL) and concentrated HCl (0.5 mL) was added. The reaction mixture was heated at reflux overnight. The reaction was cooled to RT and then 20% NaOH solution was added until pH = 13. The formed participate was filtered and washed several times with water. Compound **4e** was additionally purified by recrystallization from MeOH.

2-hydroxymethylo-1*H*-benzimidazole (**4a**) Yield = 85%. **^1^**H-NMR (500 MHz, d-DMSO) δ 4.70 (2H, s, C*H*_2_), 5.69 (1H, s br, O*H*), 7.11-7.15 (2H, m, Ar*H*), 7.49 (2H, s br, Ar*H*), 12.30 (1H, s br, N*H*); **^13^**C-NMR (125 MHz, d-DMSO) δ 57.74, 121.25, 155.02; **HRMS** calcd. for C_8_H_9_N_2_O [M + H]^+^: 149.07094, found: 149.07089.

5,6-dichloro-2-hydroxymethylo-1*H*-benzimidazole (**4c**) Yield = 84%. **^1^**H-NMR (500 MHz, d-DMSO) δ 4.70 (2H, s, C*H*_2_), 5.81 (1H, s br, O*H*), 7.73 (2H, s br, Ar*H*), 12.04 (1H, s br, N*H*); **^13^**C-NMR (125 MHz, d-DMSO) δ 57.57, 123.71, 158.10; **HRMS** calcd. for C_8_H_7_Cl_2_N_2_O [M + H]^+^: 216.99299, found: 216.99308.

5,6-dibromo-2-hydroxymethylo-1*H*-benzimidazole (**4d**) Yield = 81%. **^1^**H-NMR (500 MHz, d-DMSO) δ 4.70 (2H, s, C*H*_2_), 5.83 (1H, s br, O*H*), 7.88 (2H, s br, Ar*H*); **^13^**C-NMR (125 MHz, d-DMSO) δ 57.50, 115.47, 119.18, 157.91; **HRMS** calcd. for C_8_H_7_Br_2_N_2_O [M + H]^+^: 304.89196, found: 304.89222.

5,6-diiodo-2-hydroxymethylo-1*H*-benzimidazole (**4e**) Yield = 66%. **^1^**H-NMR (500 MHz, d-DMSO) δ 4.68 (2H, s, C*H*_2_), 5.80 (1H, s br, O*H*), 8.07 (2H, s br, Ar*H*), 12.52 (1H, s br, N*H*); **^13^**C-NMR (125 MHz, d-DMSO) δ 57.48, 98.13, 157.17; **HRMS** calcd. for C_8_H_7_I_2_N_2_O [M + H]^+^: 400.86423, found: 400.86446.


*Synthesis of 1H-benzo[d]imidazol-2(3H)-one derivatives (**5a**,**c–e**)*


Benzene-1,2-diamine derivative (3 mmol) and *N*,*N*′-carbonyldiimidazole (CDI) (3 mmol, 486 mg) were mixed in DMF (4 mL) overnight at RT. The reaction was quenched by the addition of water (20 mL). The formed participate was filtered and washed several times with water. Compound **5d** was additionally purified by recrystallization from ethyl acetate, then MeOH.

1*H*-benzo[d]imidazol-2(3*H*)-one (**5a**) Yield = 45%. **^1^**H-NMR (500 MHz, d-DMSO) δ 6.91 (4H, s br, Ar*H*), 10.56 (2H, s br, N*H*); **^13^**C-NMR (125 MHz, d-DMSO) δ 108.46, 120.40, 129.65, 155.25; **HRMS** calcd. for C_7_H_7_N_2_O [M + H]^+^: 135.05529, found: 135.05526.

5,6-dichloro-1*H*-benzo[d]imidazol-2(3*H*)-one (**5c**) Yield = 67%. **^1^**H-NMR (500 MHz, d-DMSO) δ 7.08 (2H, s br, Ar*H*), 10.91 (2H, s br, N*H*); **^13^**C-NMR (100 MHz, d-DMSO) δ 109.73, 122.35, 129.83, 155.18; **HRMS** calcd. for C_7_H_5_Cl_2_N_2_O [M + H]^+^: 202.97734, found: 202.97743.

5,6-dibromo-1*H*-benzo[d]imidazol-2(3*H*)-one (**5d**) Yield = 67%. **^1^**H-NMR (500 MHz, d-DMSO) δ 7.20 (2H, s br, Ar*H*), 10.90 (2H, s br, N*H*); **^13^**C-NMR (100 MHz, d-DMSO) δ 112.66, 113.88, 130.63, 154.93; **HRMS** calcd. for C_7_H_5_Br_2_N_2_O [M + H]^+^: 290.87631, found: 290.87626.

5,6-diiodo-1*H*-benzo[d]imidazol-2(3*H*)-one (**5e**) Yield = 67%. **^1^**H-NMR (500 MHz, d-DMSO) δ 7.41 (2H, s br, Ar*H*), 10.80 (2H, s br, N*H*); **^13^**C-NMR (100 MHz, d-DMSO) δ 96.40, 118.00, 131.46, 154.58; **HRMS** calcd. for C_7_H_5_I_2_N_2_O [M + H]^+^: 386.84858, found: 386.84861.


*Synthesis of quinoxaline derivatives (**6a**,**c–e**)*


Benzene-1,2-diamine derivative (3 mmol) and 40% glyoxal solution in water (9 mmol, 1.3 mL) were heated at reflux in EtOH (6 mL) overnight. The reaction was quenched by the addition of water (50 mL). The formed participate was filtered and washed several times with water (compound **6a** was extracted with AcOEt (3 × 30 mL). Compounds **6c** and **6d** were purified by recrystallization from MeOH. Compounds **6a** and **6e** were purified by column chromatography on silica gel, using 8:2 (hexane:AcOEt) as eluent.

quinoxaline (**6a**) Yield = 70%. **^1^**H-NMR (500 MHz, d-DMSO) δ 7.84–7.92 (2H, m, Ar*H*), 8.08–8.16 (2H, m, Ar*H*), 8.96 (2H, s br, Ar*H*); **^13^**C-NMR (125 MHz, d-DMSO) δ 129.14, 130.21, 142.23, 145.72; **HRMS** calcd. for C_8_H_7_N_2_ [M + H]^+^: 131.06037, found: 131.06042.

6,7-dichloroquinoxaline (**6c**) Yield = 99%. **^1^**H-NMR (500 MHz, d-DMSO) δ 8.46 (2H, s, Ar*H*), 9.02 (2H, s, Ar*H*); **^13^**C-NMR (125 MHz, d-DMSO) δ 130.13, 133.01, 141.15, 147.13; **HRMS** calcd. for C_8_H_5_Cl_2_N_2_ [M + H]^+^: 198.98234, found: 198.98252.

6,7-dibromoquinoxaline (**6d**) Yield = 89%. **^1^**H-NMR (500 MHz, d-DMSO) δ 8.56 (2H, s, Ar*H*), 9.01 (2H, s, Ar*H*); **^13^**C-NMR (125 MHz, d-DMSO) δ 125.77, 133.31, 141.56, 147.14; **HRMS** calcd. for C_8_H_5_Br_2_N_2_ [M + H]^+^: 286.88140, found: 286.88165.

6,7-diiodoquinoxaline (**6e**) Yield = 25%. **^1^**H-NMR (500 MHz, d-DMSO) δ 8.68 (2H, s, Ar*H*), 8.96 (2H, s, Ar*H*); **^13^**C-NMR (125 MHz, d-DMSO) δ 111.06, 138.74, 141.92, 1467.83; **HRMS** calcd. for C_8_H_5_I_2_N_2_ [M + H]^+^: 382.85366, found: 382.85426.


*Synthesis of quinoxaline-2,3-diol derivatives (**7a**–**e**)*


Benzene-1,2-diamine derivative (3 mmol) and oxalic acid (9 mmol, 810 mg) were heated at 120 °C in DMF (1 mL) overnight. The reaction was quenched by the addition of water (20 mL). The formed participate was filtered and washed several times with water (compound **6a** was extracted with AcOEt (3 × 10 mL)). Crude products were purified by recrystallization from EtOH.

quinoxaline-1,3-diol (**7a**) Yield = 57%. **^1^**H-NMR (500 MHz, d-DMSO) δ 7.02–7.10 (2H, m, Ar*H*), 7.10–7.16 (2H, m, Ar*H*), 11.90 (2H, s, 2xO*H*); **^13^**C-NMR (125 MHz, d-DMSO) δ 115.12, 123.00, 125.60, 155.17; **HRMS** calcd. for C_8_H_7_N_2_O_2_ [M + H]^+^: 163.05020, found: 163.05011.

6,7-difluoroquinoxaline-1,3-diol (**7b**) Yield = 68%. **^1^**H-NMR (500 MHz, d-DMSO) δ 7.00–7.11 (2H, m, Ar*H*), 11.92 (2H, s2xO*H*); **^13^**C-NMR (125 MHz, d-DMSO) δ 103.63, 103.74, 103.81, 122.26, 122.32, 143.91, 144.04, 145.83, 145.96, 154.70; **HRMS** calcd. for C_8_H_5_F_2_N_2_O_2_ [M + H]^+^: 199.03136, found: 199.03138.**6,7-dichloroquinoxaline-1,3-diol** (**7c**) Yield = 86%. **^1^**H-NMR (500 MHz, d-DMSO) δ 7.21 (2H, s, Ar*H*), 12.00 (2H, s, 2xO*H*); **^13^**C-NMR (100 MHz, d-DMSO) δ 115.99, 124.29, 126.03, 154.71; **HRMS** calcd. for C_8_H_5_Cl_2_N_2_O_2_ [M + H]^+^: 230.97226, found: 230.97218.

6,7-dibromooquinoxaline-1,3-diol (**7d**) Yield = 91%. **^1^**H-NMR (500 MHz, d-DMSO) δ 7.38 (2H, s, Ar*H*), 11.99 (2H, s, 2xO*H*); **^13^**C-NMR (100 MHz, d-DMSO) δ 116.15, 118.96, 126.64, 154.73; **HRMS** calcd. for C_8_H_5_Br_2_N_2_O_2_ [M + H]^+^: 318.87123, found: 318.87105.

6,7-diiodooquinoxaline-1,3-diol (**7e**) Yield = 63%. **^1^**H-NMR (400 MHz, d-DMSO) δ 7.51 (2H, s, Ar*H*), 11.77 (2H, s br, 2xO*H*); **^13^**C-NMR (100 MHz, d-DMSO) δ 99.11, 124.80, 128.06, 155.49; **HRMS** calcd. for C_8_H_5_I_2_N_2_O_2_ [M + H]^+^: 414.84349, found: 414.84366.

### 4.2. NanoDSF

hCK2α was expressed and purified according to the method described previously [35]. Thermal stability of 1:10 protein–ligand complexes was monitored on Prometheus (Nanotemper Technologies, München, Germany) using PR-C002 Prometheus NT.48 Standard Capillaries using temperature ramp of 20 to 80 °C with 1 °C/min heating rate. Temperature-induced variation in ratio fluorescence intensity at 330 and 350 nm was globally analyzed using the procedure described before [37]. All experiments were performed in 25 mM Tris–HCl (pH 8, 0.5 M NaCl) with 2% (*v/v*) DMSO. Protein was kept in constant 2.5 μM concentration.

### 4.3. Inhibitory Activity against Human CK2α

The inhibitory effect of 8 selected compounds was monitored based on the luminescence measured with the SpectraMax iD3 Multi-Mode Microplate Reader (Molecular Devices, San Jose, CA, USA), using the previously described method with the aid of ADP-Glo kinase assay (Promega, Walldorf, Germany) [43]. Inhibitor concentration varied in the range of 1 nM–1 mM (see Appendix A). Each experiment was repeated in triplicate and IC_50_ values were further estimated globally using the sigmoidal dose-response equation implemented in Origin 9.4 package (www.originlab.com, upgraded 20.07.27).

### 4.4. Cell Viability Assay

The following human cell lines were used in this study: (i) epidermoid carcinoma A-431 (ATCC^®^ CRL-1555), (ii) colorectal carcinoma HCT116 (ATCC^®^ CCL-247), (iii) colorectal carcinoma HCT116p53^−/−^, and (iv) normal fibroblasts BJ (ATCC^®^ CRL-2522). The HCT116p53^−/−^ cell line was a kind gift from professor Bert Vogelstein (John Hopkin’s University, Baltimore, MD, USA). The remaining cell lines were purchased from ATCC (Manassas, VA, USA) or Sigma Aldrich by Merck (Darmstadt, Germany).

Cells were cultured in Dulbecco’s modified Eagle’s or Roswell Park Memorial Institute (RPMI)-1640 medium supplemented with 10% or 15% fetal bovine serum (Gibco by Thermo Fisher Scientific, Waltham, MA, USA) under standard culture conditions (37 °C, 5% CO_2_) in a humidified incubator with atmospheric oxygen.

The viability of cells was assayed by measuring the conversion of MTT (3-(4,5-dimethylthiazol-2-yl)-2,5-diphenyltetrazolium bromide) to the formazan (the rate of this reaction is proportional to the number of surviving cells). Cells were seeded in a 96-well plate at a density of 1500–2500 cells per well, 24 h before treatment. Treatment with tested compounds and cisplatin (0.25–160 µM) was performed for 72 h. MTT stock solution (Sigma-Aldrich) was added at the final concentration 0.5 mg/mL. After 2 h of incubation at 37 °C, water-insoluble formazan was dissolved in a lysis buffer containing 20% SDS, 50% DMF, 2.5% hydrochloric acid, and 2.5% acetic acid. Optical densities were measured at 570 nm using a scanning multi-well spectrophotometer (PARADIGM Detection Platform; Beckman Coulter, Brea, CA, USA).

### 4.5. Chromatographic Hydrophobicity Index

Chromatographic hydrophobicity index (log(τ)) was determined with the reverse phase HPLC (RP-HPLC). Column: Nova-pak^®^ C18 4 μm, 4.6 × 250 mm. Eluent: 2:1 methanol: 20 mM aqueous ammonium formate (pH = 8) (*v/v*). Flow: 0.7 mL/min. Detector wavelength: 280 nm. The individual retention times (T_ret_) were converted to a hydrophobicity scale according to the formula τ = (T_ret_ − T_0_)/T_0_. where the retention time of unretained solvent (T_0_) was estimated to 300 s [24].

### 4.6. Molecular Modeling

Docking of all tested ligands at the binding site of the catalytic subunit of human protein kinase CK2 was performed with VINA algorithm implemented in Yasara Structure package (www.yasara.com, version 20.12.24) using amber14 force-field with 8 Å cutoff for long-range electrostatic interactions. We used 8 just-resolved structures of complexes of hCK2α with variously brominated benzotriazoles (Protein Data Bank records 6TLW, 6TLV, 6TLU, 6TLS, 6TLR, 6TLP, 6TLO, and 6TLL) as the reference protein structures. This, including alternative protein conformations, stated 18 structures to be tested as the ‘receptor’. Such an extended set of template protein structures was chosen to allow the sampling of a larger conformational space that allows multiple ligand orientations at the binding site (see Appendix A). In silico screening procedure was restricted to the cuboid region that contained all residues from the ATP binding site, further additionally extended by 2 Å in each direction. During simulations, the sidechain atoms of protein residues proximal to the location of the original ligand (4 Å threshold) were flexible, while coordinates of all other protein atoms were kept fixed. Finally, each of 35 ligands was subjected to 25 independent cycles of the docking procedure using 18 structural templates. The resulting structures were then clustered, and for each ligand, the highest scored one was used as the representative. This score was used as a rough estimate of the free energy of ligand binding.

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
