# Peer review of "Synthesis of Novel Halogenated Heterocycles Based on o-Phenylenediamine and Their Interactions with the Catalytic Subunit of Protein Kinase CK2"

_molecules, 2021, doi:10.3390/molecules26113163_

Round 1

Reviewer 1 Report

I have no objection and accept the paper in present form, but

there is some minor correction in text editing needed eg.

page 5 lines 131  ... the effect of halogen substitution in the most hydrophobic 2,1,3-Benzothiadiazole ...

page 9 line 259   ...some of the tested here2-Trifluoromethyl-1H-benzimidazole derivatives.... should be written with a lowercase letter.

Author Response

Numerous minor corrections have been done.

Reviewer 2 Report

The study presented in this manuscript leads to potentially interesting CK2 inhibitors. A table with the IC50 values of the inhibition of the enzyme by the compounds must be added. Moreover, the activity of at least one derivative on a sensitive cancer cell line must be performed, to confirm that the study effectively identified good CK2 inhibitors.

Author Response

The whole text has been carrefully edited.

We have determined inhibitory activities for eight most promissing compounds.

We have additionally made toxicity assay for the reference cell lines (both cancer and normal).

All these data are summarized in Table 2, while the analysis is shown in suplemantary figures  S30-S38.

Reviewer 3 Report

The article “Synthesis of novel halogenated heterocycles based on o-phenylenediamine and their interactions with the catalytic subunit 4 of protein kinase CK2” presents a series of novel halogenated derivatives against CK2. Overall, the article is not easy to follow and of poor quality. In the results, the authors state the obvious, that is that halogen atoms are hydrophobic and that the increase in hydrophobicity increases binding, which is known as hydrophobic effect. Additionally, the method the authors use to measure de binding affinity is semi-qualitative and they use it to compare the results with the predicted binding affinity (that would be the binding enthalpy) obtained with molecular modeling techniques. The authors have only considered for modelling purposes a series of PDBs that have been deposited in the PDB by two of the signing authors of this paper, and have not taken into consideration any of the other previous crystal structures of CK2 bound to TBB or TBB-derived analogues. Moreover, the docking calculations of the series of compounds that clearly will have to establish halogen bonds with the ATP-binding site of CK2 have been carried out using software that does not have any parametrization for the halogen bonds. If the molecular modeling programs do not take into account the driving interaction of the binding, the resulting binding mode is not going to be accurate. Additionally, Figure 5 (that is not referenced to in the text) is completely unintelligible and fails to explain the interactions with the ATP binding site as stated in the title of this article.

Author Response

Additionally, the method the authors use to measure de binding affinity is semi-qualitative and they use it to compare the results with the predicted binding affinity (that would be the binding enthalpy) obtained with molecular modeling techniques. Additionally, the method the authors use to measure de binding affinity is semi-qualitative and they use it to compare the results with the predicted binding affinity (that would be the binding enthalpy) obtained with molecular modeling techniques. The authors have only considered for modelling purposes a series of PDBs that have been deposited in the PDB by two of the signing authors of this paper, and have not taken into consideration any of the other previous crystal structures of CK2 bound to TBB or TBB-derived analogues. Moreover, the docking calculations of the series of compounds that clearly will have to establish halogen bonds with the ATP-binding site of CK2 have been carried out using software that does not have any parametrization for the halogen bonds. If the molecular modeling programs do not take into account the driving interaction of the binding, the resulting binding mode is not going to be accurate. Additionally, Figure 5 (that is not referenced to in the text) is completely unintelligible and fails to explain the interactions with the ATP binding site as stated in the title of this article.

We have used Molecular Modeling to mainly select the scaffolds than be accommodated at the ATP-binding site. This is now explicitaly written in the text.

The free energy (or enthalpy) values derived from MM methods are generally semi-quantitative, but we decide to show the because at the moment we have no crystall structures of the complexes.

We have used MM as the screening procedure that preceded experimental screening based on nanoDSF. Following the Reviewers comments we have performed direct measurements of inhibitory activity and toxicity of some ligands that caused the largest thermal shift in nanoDSF.
We decide to show all these experiments to demonstate how efficient are virtual and nanoDF screenings. 

Please note, that comonly a very limited number of "receptor" conformers are used. Contrary to common approach, we used as protein model over a dozen of structures determined recently in our groups. We chose our own ones because all of them were crystalized in the identical conditions and were refined using identical protocols. Moreover, the ligand poses sample the whole space of the ATP-binding sites, so we dod not expect tempate-induced preferences towars a particular conformation. This is now explained in the manuscript. We have prepared new Figures S39-S41, the old one is removed.

Reviewer 4 Report

The paper reports the synthesis of six series of halogenated heterocyclic compounds (4 to 5 compounds of each series), and evaluation of their interactions with CK2 kinase. Thirty-two compounds were synthesized and their logDs determined. Binding affinities of the synthesized compounds towards hCK2α were assessed/estimated using nano-DSF and molecular modeling. The results of the semi-qualitative measure of the binding affinity showed that most compounds interact weakly with hCK2α. Molecular modeling was used to test that the thermodynamic contribution of non-specific hydrophobic interactions of halogens to the free energy of ligand binding was independent from direct electrostatic interactions involving more polar heterocyclic rings. The innovation is rather limited, as several of the synthesized compounds are already reported as CK2 inhibitors, and some of the conclusions obtained with these studies are already described in the literature. Nevertheless, in this work the authors conclude that quinoxaline-2,3-diol and 2-hydroxymethyl-1H-benzimidazole derivatives might lead to novel potent CK2 inhibitors. To support this conclusion, at least the IC50 values of 4e and 7e should be determined and compared with compound 3e.

Other details:

- the English should be revised (for example, too much unnecessary “the”)

- Scheme 1, the yields obtained should be added in the scheme or in the legend, and included in the discussion of section 2.1

- at least, a general protocol for the synthesis of each series of compounds, as well as the detailed data and spectral data of the two most promising compounds, should be included in section 4.1 instead of being available in the supplementary materials.

Author Response

We have determined inhibitory activitues for eight compounds, including 3e, 4e, 7e. 3e was found more active than 3d, and 5e was even stronger inhibitor.

We have carrefuly revised the manuscript

The whole protocols, including yields are now included in the main text.

Reviewer 5 Report

The manuscript entitled “Synthesis of novel halogenated heterocycles based on o-phe-2 nylenediamine and their interactions with the catalytic subunit 3 of protein kinase CK2 reports the results of an investigation devoted to the design, binding measurements and molecular modeling studies of new series of heterocyclic halogenated ligands anticipated to have a significant hydrophobic character partly explaining their interactions with CK2 protein kinase. The manuscript is well written and presents original results. For these reasons, it should be accepted provided the importants points discussed below are taken into account.

Major points

  • The major point in my opinion is related to the way the effect of halogens atoms is presented and discussed since the authors focussed on the « …hydrophobic interactions directly attributed to the presence of the halogen atoms… » and their relation with the affinity of the ligands. The authors should broaden the perspective, including halogen bonding for Br, Cl and I atoms and hydrogen bonding interactions for Fluorine. In fact, in the discussion (from the first paragraph p. 7) the authors begin to mention these important properties with respect to binding and affinity but it should be introduced and discussed before (from the introduction).
  • Always in relation to this point, the discussion on the molecular modelling results is, as indicated by the authors, to be taken with caution since the tools used have not been properly parametrised to be able to describe with precision the structural and energetic properties of the ligand…receptor interactions investigated with these halogens ligands. If the question is not to restart and do again all the calculations with a more suitable methodology, an extension of the discussion of these limitations appears necessary. More precisely, the discussion at the top of page 8, lines 235-236 ; … however since the force-field used in docking procedure does not include explicit potential for a halogen bond, these interactions are underestimated (blue markers in Figure 4)… needs to be  reinforced, for example by relying on recent surveys of the PDB for halogenated ligands…receptor interactions (Shinada et col., J. Med. Chem. 2019, 62, 9341−9356) or by comparing their results with other studies of the litterature for these kinds of ligands.

Minor point

In the material and method section, in the Molecular Modelling paragraph, the resolution of the various pdb structures used should be specified.

Author Response

We have rephrased  part concerning molecular modeling, but all these analyses should be treated rather quantitatively. We have add general information concerning halogen bonding in the intrduction, just below figure 1.

According to nanoDSF and MM data shown in Figure 5 3e, 4e and 7e possibly forms halogen bonds with the hinge region ( Supplementary Fugure S40, S41).

We have reinterprated MM data according to the preferred orientation rather than energy terms (supplementary Table S1)

Round 2

Reviewer 2 Report

The article is suitable for the publication in this form

Reviewer 3 Report

The authors have addressed all the previously commented issues.

Reviewer 4 Report

The authors have replied to most of my comments. Please correct in the abstract "bining", and in the conclusions "referenence".